# Titin-Related Dilated Cardiomyopathy: The Clinical Trajectory and the Role of Circulating Biomarkers in the Clinical Assessment

**DOI:** 10.3390/diagnostics12010013

**Published:** 2021-12-22

**Authors:** Przemysław Chmielewski, Grażyna Truszkowska, Ilona Kowalik, Małgorzata Rydzanicz, Ewa Michalak, Małgorzata Sobieszczańska-Małek, Maria Franaszczyk, Piotr Stawiński, Małgorzata Stępień-Wojno, Artur Oręziak, Michał Lewandowski, Przemysław Leszek, Maria Bilińska, Tomasz Zieliński, Rafał Płoski, Zofia T. Bilińska

**Affiliations:** 1Unit for Screening Studies in Inherited Cardiovascular Diseases, National Institute of Cardiology, 04-628 Warsaw, Poland; pchmielewski@ikard.pl (P.C.); emichalak@ikard.pl (E.M.); mstepien@ikard.pl (M.S.-W.); 2Molecular Biology Laboratory, Department of Medical Biology, National Institute of Cardiology, 04-628 Warsaw, Poland; gtruszkowska@ikard.pl (G.T.); m.fran@wp.pl (M.F.); 3Clinical Research Support Center, National Institute of Cardiology, 04-628 Warsaw, Poland; ikowalik@ikard.pl; 4Department of Medical Genetics, Medical University of Warsaw, 02-106 Warsaw, Poland; mrydzanicz@wum.edu.pl (M.R.); stawinski84@gmail.com (P.S.); 5Department of Heart Failure and Transplantology, National Institute of Cardiology, 04-628 Warsaw, Poland; m.sobieszczanska@ikard.pl (M.S.-M.); pleszek@ikard.pl (P.L.); tzielin@ikard.pl (T.Z.); 6Department of Arrhythmia, National Institute of Cardiology, 04-628 Warsaw, Poland; aoreziak@ikard.pl (A.O.); mbilinska@ikard.pl (M.B.); mlewan@ikard.pl (M.L.)

**Keywords:** cardiotitinopathy, *TTN* truncating variants, troponin T, NT-proBNP, malignant ventricular arrhythmia, end-stage heart failure

## Abstract

Titin truncating variants (*TTN*tv) are known as the leading cause of inherited dilated cardiomyopathy (DCM). Nevertheless, it is unclear whether circulating cardiac biomarkers are helpful in detection and risk assessment. We sought to assess 1) early indicators of cardiotitinopathy including the serum biomarkers high-sensitivity cardiac troponin T (hs-cTnT) and N-terminal pro-B-type natriuretic peptide (NT-proBNP) in clinically stable patients, and 2) predictors of outcome among *TTN*tv carriers. Our single-center cohort consisted of 108 *TTN*tv carriers (including 70 DCM patients) from 43 families. Clinical, laboratory and follow-up data were analyzed. The earliest abnormality was left ventricular dysfunction, present in 8, 26 and 47% of patients in the second, third and fourth decade of life, respectively. It was followed by symptoms of heart failure, linked to NT-proBNP elevation and severe left ventricular systolic dysfunction, and later by arrhythmias. Hs-cTnT serum levels were increased in the late stage of the disease only. During the median follow-up of 5.2 years, both malignant ventricular arrhythmia (MVA) and end-stage heart failure (esHF) occurred in 12% of *TTN*tv carriers. In multivariable analysis, NT-proBNP level ≥650 pg/mL was the best predictor of both composite endpoints (MVA and esHF) and of MVA alone. In conclusion, echocardiographic abnormalities are the first detectable anomalies in the course of cardiotitinopathies. The assessment of circulating cardiac biomarkers is not useful in the detection of the disease onset but may be helpful in risk assessment.

## 1. Introduction

Dilated cardiomyopathy (DCM) is a major cause of heart failure (HF) and has a genetic basis in 40 to 50% of cases [1]. Titin truncating variants (*TTN*tv*)* account for as many as 20–25% of the genetic background in DCM [2,3,4,5,6,7] of European but not African ancestry [8]. *TTN*tv is also found in different forms of DCM, including peripartum [4,9], alcoholic [10], chemotherapy-induced [11] and arrhythmogenic DCM. Earlier studies showed good response to optimal medical therapy [7,12,13], the impact of mutation location on the course of disease [2,6] and similar prognosis as in other forms of DCM [2,3]. Nevertheless, none of the studies involved circulating cardiac biomarkers in baseline characteristics. So, there are no data on their performance in the early stage of the disease or their role as markers of prognosis. The timing of the appearance of arrhythmia in cardiotitinopathies is not well characterized, either. Recently, the presence of *TTN*tv was shown to be an important risk factor for clinically significant arrhythmia in DCM patients [14,15].

In 2017, we published our data on cardiotitinopathies including characteristics of 16 *TTN*tv identified in 17 probands and their 29 relative and compared them to *TTN*tv noncarriers [3]. Data on the probands were also included in the multicenter study by Akhtar et al. [16]. Since then, we have identified 29 *TTN*-related DCM probands and their 39 *TTN*tv-positive relatives. Recently, we showed that elevated high-sensitivity cardiac troponin T (hs-cTnT) serum concentration is the earliest cardiomyopathy indicator in a cohort of DCM-causing lamin A/C gene (*LMNA*) mutation carriers [17], preceding arrhythmias, conduction defects and left ventricular systolic dysfunction (LVSD). Increased levels of hs-cTnT and the N-terminal pro-B-type natriuretic peptide (NT-proBNP) were also the strongest risk factors of malignant ventricular arrhythmia (MVA) occurrence in that study cohort. As these findings may substantially facilitate the care of *LMNA* mutation carriers, we wanted to check whether they could be repeated in cardiotitinopathies, the most common form of inherited DCM. Therefore, we sought to assess the clinical characteristics including serum biomarkers, the penetrance of abnormal clinical findings and prognostic risk factors in our cohort of *TTN*tv carriers.

## 2. Materials and Methods

### 2.1. Study Design

The study cohort consists of carriers of DCM-causing *TTN*tv who were identified in the National Institute of Cardiology, Warsaw as a result of genetic testing offered to all DCM probands in the care of our Unit and subsequent cascade screening offered to all probands’ families. All *TTN*tv were identified between 2012 and 2021 and were considered pathogenic or likely pathogenic according to the American College of Medical Genetics and Genomics (ACMG) criteria [16,18].

Medical data of all probands and relatives were retrospectively collected, including baseline clinical information from the first documented visit to the Institute, prior medical records and follow-up data. We analyzed the baseline data, comprising medical history, clinical examination, 12-lead electrocardiography, two-dimensional Doppler echocardiography, 24-h Holter ECG monitoring, as well as the serum biomarkers hs-cTnT and NT-proBNP, measured during ambulatory visits in patients in a stable condition. In all probands, coronary computed tomography angiography or coronary angiography was performed. All records were reviewed for the first documented occurrence of disease indicators such as echocardiographic anomalies, HF symptoms, arrhythmias, conduction defects and elevation of serum cardiac biomarkers, as well as for major cardiovascular events. In the case of patients whose first contact to the Institute took place during HF exacerbations, the baseline evaluation was moved to latter ambulatory visits whenever possible. When they could be assessed only in the acute phase of the disease, the data on serum biomarker levels and medical therapy were not included in the characteristics. 

We sought to examine the order of appearance of cardiotitinopathy indicators, including elevated circulating biomarkers. We also wanted to evaluate the prognostic value of circulating cardiac biomarker concentrations with regard to the occurrence of MVA and esHF during the follow-up period. 

### 2.2. Definitions

Left ventricular enlargement (LVE) was ascertained when the left ventricular end diastolic diameter (LVEDD) exceeded 112% of the predicted value, corrected for age and body surface area according to Henry’s formula, while LVSD was ascertained when the left ventricular ejection fraction (LVEF) was <50%. We used the term left ventricular dysfunction (LVD) when one of the abovementioned criteria was met, and the diagnosis of DCM was made when both criteria were met. When no LVE but more distinct LVSD was present (LVEF < 45%), hypokinetic non-dilated cardiomyopathy (HNDC) was diagnosed; for the purpose of this study, patients with HNDC were included in the DCM cohort. When LVEF fell below 35%, we used the term severe LVSD. In the presence of relevant abnormalities, not sufficient for the diagnosis of DCM/HNDC, such as LVE > 117%, LVEF 45–49%, cardiac conduction defect (CCD), atrial or ventricular arrhythmias unexplained by other conditions, we used the term indeterminate cardiomyopathy, which may represent the preclinical phase of DCM [19]. 

CCD included atrioventricular block (AVB) and left bundle branch block (LBBB). First-degree AVB was defined by a PR interval >200 ms on standard 12-lead ECG. High-degree AVB included type II second-degree or third-degree AVB. Atrial arrhythmias (AA) included atrial fibrillation (AF), flutter and paroxysmal atrial tachycardia lasting ≥30 s. Ventricular arrhythmias (VA) included a ventricular ectopy burden of >500/24 h or non-sustained ventricular tachycardia (nsVT), defined as ≥3 consecutive ventricular beats at >120 bpm on Holter monitoring. If the VT lasted over 30 sec., it was considered sustained (sVT).

HF was recognized in the presence of typical symptoms, accompanied by structural and/or functional cardiac abnormalities, resulting in reduced cardiac output and/or elevated intracardiac pressures. The symptoms of HF were assessed using New York Heart Association classification (NYHA classes 1–4). The HF condition was considered stable when there had been no worsening of HF for ≥3 months. When the symptoms of HF had acute onset, were refractory to medical therapy and led directly to heart transplantation (HTx) or the implantation of a left ventricular assist device (LVAD), we used the term fulminant HF. End-stage HF was defined as HTx, LVAD implantation or death caused by HF.

MVA was defined as sudden cardiac death (SCD), cardiopulmonary resuscitation (CPR) or appropriate implantable cardioverter defibrillator (ICD) intervention, i.e., an ICD discharge or anti-tachycardia pacing (ATP) for termination of ventricular fibrillation/VT. Death was classified as sudden if it occurred within 1 h of the onset of cardiac manifestations, or during sleep (in the absence of previous hemodynamic deterioration), or within 24 h after the patient was last seen apparently stable clinically. Sudden cardiac arrest (SCA) was defined as occurring within 1 h of the onset of acute symptoms and reversed by CPR. 

Relatives included all probands’ family members with *TTN*tv identified as a result of cascade screening, irrespective of the degree of kinship. A family history of SCD was considered positive if ≥1 first-degree relative had died suddenly before the age of 50 years. 

### 2.3. Biomarker Measurements

The plasma levels of NT-proBNP were measured by the electrochemiluminescent immunoassays Elecsys 2010 (Roche, Mannheim, Germany) with the upper limit of normal values defined by the manufacturer at 125 pg/mL. The plasma levels of cardiac troponin T were measured by the troponin T hs-STAT (Roche, Mannheim, Germany) with the upper limit of normal values defined by the manufacturer at 14 ng/L. All measurements were performed in the National Institute of Cardiology laboratory.

### 2.4. DNA Sequencing and TTN Mutation Analysis

DNA was extracted from the peripheral blood by phenol extraction, the salting out method or using the Genomic Maxi AX kit (A&A Biotechnology, Gdynia, Poland). Next-generation sequencing (NGS) was performed in 46 probands using whole exome sequencing (WES) in 27 probands, the TruSight One Sequencing Panel Kit (TSO) in 4 probands or the TruSight Cardio Sequencing Kit (TSC) in 15 probands (Illumina, San Diego, CA, USA). WES libraries were prepared using the TruSeq Exome Enrichment Kit (llumina), Nextera DNA Sample Preparation Kits (Illumina) or the Twist Human Core Exome Kit (Twist Bioscience, South San Francisco, CA, USA). TSO sequencing was performed similarly to WES using the Nextera DNA Sample Preparation Kit (Illumina) with only the difference in the enrichment probes used. WES and TSO libraries were paired-end sequenced (2 × 100 bp) on Illumina HiSeq1500 or NovaSeq 6000 and TSC libraries on Illumina MiSeqDx. Library preparation, sequencing and data analysis were performed as described previously [20]. *TTN*tv identified with NGS was followed-up in probands and relatives with Sanger sequencing using BigDye Terminator v3.1 or the v1.1 Cycle Sequencing Kit (Life Technologies, Carlsbad, CA, USA) according to the manufacturer’s instructions and the 3500xL or 3130xL Genetic Analyzer (Life Technologies). The results were analyzed with Variant Reporter 1.1 Software (Life Technologies). Other genes with strong or moderate evidence of causative relationship with DCM (*ACTC1, ACTN2, BAG3, DES, DSP, FLNC, JPH2, LMNA, MYH7, NEXN, PLN, RBM20, SCN5A, TNNC1, TNNI3, TNNT2, TPM1, VCL)* [21] were also inspected for the presence of rare variants. The identified variants were classified according to ACMG criteria [18,22]. Carriers of pathogenic or likely pathogenic variants in other DCM-related genes were excluded from the study.

### 2.5. Statistical Analysis

All results for categorical variables were presented as counts and percentages and for continuous variables as the mean and standard deviation (SD) or median and quartiles (Q1:25th, Q2:75th percentiles). The chi-square independence or Fisher exact test was used for the comparison of categorical variables. The differences between continuous variables were tested by Student’s *t*-test (for two independent samples, normally distributed data), or in the case of a skewed distribution, non-parametric Mann–Whitney tests. In order to estimate intra-family relationships, intra-class correlation coefficients were calculated for the NT-proBNP and the hs-cTnT measurements (after log transformation). 

Penetrations were calculated using the maximum likelihood estimator—the Kaplan–Meier method. Patients were excluded from the analyses when we had no data on test results used for the detection of individual abnormalities, e.g., Holter recordings for VA detection or serum biomarker measurements for hs-cTnT or NT-proBNP elevation. The first documented occurrence of a prespecified abnormality was considered an event. In subjects without an event, the follow-up period extended to the most recent evaluation before 31 May 2021.

A receiver-operating characteristic curve (ROC) analysis was used to assess the cut-off point of the markers for the prediction of events. The optimal cut off was defined as the value with the maximal sum of sensitivity and specificity. Event analysis over time was conducted using the univariable and multivariable Cox proportional-hazards regression model. In order to indicate independent predictors of events, the stepwise (backward) variable selection procedure was used. All variables with a significant prognostic impact in the univariable analysis (*p* ≤ 0.10) were included in the multivariable model. Then a backward selection was used to create the final model. The proportionality of hazards was verified using weighted Schoenfeld residuals. Model discrimination was assessed using Harrell’s Concordance Statistics (C-index). Risk was quantified as a hazard ratio (HR) with a 95% confidence interval (CI). In the case of zero events in one of the subgroups, HRs were calculated using Firth’s penalized likelihood approach. Survival curves were constructed by the Kaplan–Meier method and compared by the log-rank test. All hypotheses were two-tailed with 0.05 type I error. All statistical analyses were performed using SAS statistical software, version 9.4 (SAS Institute, Cary, NC, USA).

## 3. Results

### 3.1. Molecular Findings in the Study Cohort

We identified 41 different *TTN*tv in 46 unrelated probands: Two variants were shared by two probands each and one was identified in four probands. As a result of analysis of other DCM-related genes, likely pathogenic variants were identified in *MYH7*, *SCN5A* and *TNNT2* in three probands. Of the 41 *TTN*tv, 30 had been described before in ClinVar, Varsome or HGMD databases and 11 are novel. The identified variants are pathogenic (n = 40) or likely pathogenic (n = 1) according to ACMG criteria. Two of them are located in the Z-disc (both nonsense), 10 in the I-band (5 nonsense, 4 frameshift and 1 splice-site), 28 in the A-band (12 nonsense and 16 frameshift) and one in the M-band (nonsense). As a result of genetic screening in the families, 68 *TTN*tv carriers were identified among relatives. Three probands with likely pathogenic variants in other DCM-related genes and their two relatives, as well as a patient after heart transplantation (HTx) performed at another center, were excluded from the study. The details of the identified variants in *TTN* and other DCM-related genes are shown in Appendix A. 

### 3.2. Clinical Characteristics of the Study Population

The study cohort was composed of 108 subjects: 70 DCM patients (all 43 probands and 27 relatives), 13 subjects with other cardiac abnormalities, labelled as indeterminate cardiomyopathy, and 25 healthy relatives with no signs of cardiomyopathy. Four patients experienced fulminant HF at the onset of their disease; they all underwent left ventricular assist device (LVAD) implantation: As a bridge to urgent HTx in three cases and as a bridge to recovery in one case. The baseline clinical characteristics of 108 TTNtv carriers are given in the Table 1 and in Appendix A. 

The DCM patients (pts) were young (mean age 40 years), the majority of them were male (79%) and showed features of mild HF at the initial visit: 80% were in NYHA class 1–2, the mean LVEF was 36% and the median NT-proBNP concentration was 534 pg/mL. Atrial arrhythmias were found in 31%, nsVT in 55% and CCD in 29% of them.

We found no significant difference in age between them and their non-DCM relatives, suggesting incomplete penetrance and a mild course of cardiotitinopathy in some carriers, especially in women who made up the majority (68%) of the non-DCM group. Arrhythmias and CCD were infrequent and could be found in a total of 18% of them. 

The hs-cTnT concentration measured in a stable condition was higher in DCM vs. non-DCM patients but it remained low (median 6.7 vs. <3.0 ng/L, respectively). The intra-family correlation for NT-proBNP and hs-cTnT levels was absent (the correlation for the logarithm of NT-proBNP concentration was −0.01 and the logarithm of hs-cTnT was 0.11).

Of note, sudden cardiac death (SCD) under the age of 50 years in family history was found in 26% of patients.

### 3.3. Penetrance of Cardiotitinopathy Indicators

Penetrance of cardiac abnormalities in the course of cardiotitinopathy was age-dependent (Figure 1). The earliest abnormality was left ventricular dysfunction (LVD), defined as LVEF < 50% or left ventricular enlargement (LVE) > 112%. It was detected in the second, third and fourth decade of life in 8%, 26% and 47% of carriers, respectively. It anticipated the onset of HF symptoms by 5–10 years, accompanied by NT-proBNP elevation, transient or persistent severe LVSD and VA (Appendix A). AA and AVB appeared late, preceding the occurrence of such adverse events as MVA and esHF. An elevated hs-cTnT concentration seems to be an indicator of the end-stage phase of cardiotitinopathy. 

### 3.4. Results of Screening in Carriers of Cardiotitinopathy-Causing Truncating Variants

Observations on the sequence of occurrence of individual abnormalities may be biased due to the fact that the disease is often detected at an advanced stage with a number of anomalies already present. Therefore, we analyzed the penetrance of cardiac abnormalities also in the group of 49 *TTN*tv carriers who came to our Unit for screening. The diagnosis of DCM was established in 11 of them, however only one patient developed HF during the follow-up. In this group, LVD was detected as the earliest indicator of *TTN*tv carriership in 21 (43%) pts (Figure 2). Of note, in 16% of pts, LVE was accompanied by LVSD whereas LVE and LVSD were found as isolated deviations in 16% and 10% of pts, respectively. NT-proBNP serum concentration was elevated in only 28% of pts with LVD.

Arrhythmias and CCD were detected less frequently than in the whole cohort, despite repeated ECG and Holter recordings. Interestingly, AA was detected in four (8%) pts, and they preceded LVD in all of them. Furthermore, VA in two cases and AVB in one case were detected in subjects without LVD. 

### 3.5. Outcome and Risk Stratification in Cardiotitinopathy

The median follow-up in the group of 108 *TTN*tv carriers was 5.2 years [Q1: 2.1, Q3: 7.9]. During the follow-up period, 13 (12%) patients developed esHF (Table 2): Five (5%) of these pts died of HF and eight (7%) were transplanted (in five cases, preceded by LVAD implantation). MVA, mostly adequate ICD interventions, also occurred in 13 (12%) pts (Table 2).

We examined the influence of pre-specified risk factors on the risk of occurrence of composite endpoints (esHF or MVA) in the whole cohort of 108 *TTN*tv carriers (Table 3). The univariable analysis suggests an impact of such factors, such as severely reduced LVEF, dilated left atrium, elevated NT-proBNP and hs-cTnT, the presence of left bundle branch block (LBBB), non-sustained ventricular tachycardia (nsVT) or AA. In multivariable analysis, NT-proBNP level ≥650 pg/mL was the best predictor of the composite endpoint at 6 years of follow-up (Figure 3). The model had good discrimination as evidenced by the C-index of 0.842 [95% CI: 0.776–0.908].

Anticipating life-threatening arrhythmia episodes is even more important as they can be interrupted by ICD interventions. The univariable analysis of MVA events during follow-up in the group of 107 patients with no history of SCA or sVT shows the possible influence of such risk factors such as severely reduced LVEF, dilated left atrium, elevated NT-proBNP, the presence of LBBB, AA or nsVT but no impact of sex, family history of SCD or elevated hs-cTnT (Table 4). In multivariable analysis, NT-proBNP level ≥650 pg/mL was again the best predictor of MVA at 6 years of follow-up (Figure 4). The model had good discrimination as evidenced by the C-index value of 0.787 [95% CI: 0.672–0.909].

## 4. Discussion

### 4.1. Penetrance of Cardiotitinopathy Indicators

A major finding of the study is that the earliest marker of the carrier status in *TTN*tv-related DCM is left ventricular dysfunction defined as LVEF < 50% or LVE > 112%. It preceded the development of overt HF and severe LVSD by 5–10 years, which, in turn, was followed by a variety of arrhythmia, both ventricular and atrial, as well as AVB. Of note, among screened relatives (n = 49), isolated LVE with normal LVEF was the first sign of cardiotitinopathy in 16% of subjects, reduced LVEF without LVE in 10% and both abnormalities were detected simultaneously in 16% of subjects. LVE is known as the first sign of early DCM [23,24,25]. *TTN*tv was also associated with the eccentric cardiac remodeling in the analysis of cardiac magnetic resonance in healthy humans [26]. This is in agreement with a proposal of a new definition of DCM, which recognizes three forms of the preclinical phase of DCM, including isolated left ventricular dilation [19]. However, in a *ttna*tv/+ vs. *ttna*+/+ zebrafish model, serial echocardiography showed significant LVEF reduction preceding LVE by 3–6 months [27]. Furthermore, comprehensive genomics-first studies on the impact of *TTN*tv on the cardiac phenotype by Haggerty et al. [8] and Pirruccello et al. [28] showed that *TTN*tv carriers are characterized by lower LVEF but not larger left ventricular (LV) diastolic dimensions or volumes. Our *TTN*tv-related DCM-dedicated study shows that both LVE and LVSD may be the first detectable abnormality.

To the best of our knowledge, circulating cardiac biomarkers in relation to either disease penetrance or prognosis have not been reported in cardiotitinopathies. The role of circulating cardiac biomarkers in the detection of HF is widely recognized [29]. In community-based studies, multiple cardiac biomarkers are detectable in ambulatory individuals and add prognostic value to standard risk factors for predicting mortality, overall cardiovascular events and HF [30,31]. However, little is known about the significance of circulating biomarkers in the early stage of DCM in humans [30,32]. We recently showed that elevated hs-cTnT is the earliest marker of the carrier status in cardiolaminopathies [17] and might be a “red flag” in asymptomatic or mildly symptomatic carriers.

Our data show that measurements of widely available serum biomarkers cannot replace echocardiography in the detection of affected *TTN*tv carriers. NT-proBNP serum level is rarely elevated in subjects with mildly reduced LVEF or isolated LVE and it usually exceeds the normal range when HF symptoms and advanced LVSD are present. In contrast to cardiolaminopathies, an elevated hs-cTnT concentration seems to be an indicator of the end-stage phase of cardiotitinopathies.

In our cohort, AA was found during baseline evaluation in 23% of *TTN*tv carriers and VA in 40%. The frequencies are lower than in the largest study to date by Akhtar et al. who found AA in one-third and VA in one-half of *TTN*tv carriers [16]. The difference can be explained by the younger population and less-advanced LVSD in our study. The timing of the appearance of VA seems similar in both cohorts, in conjunction with LVSD progression. 

Authors of several recent papers report a significant arrhythmic burden characteristic of *TTN*tv-associated DCM, most often found in a relatively advanced disease stage. In the study by Corden et al., having a *TTN*tv was associated with a higher risk of receiving appropriate ICD therapy in the group of 148 DCM patients with implanted ICDs [15]. In addition, *TTN*tv was a risk factor for developing new persistent AF [15]. Tayal et al. found that patients with *TTN*tv are more likely to have a history of AA or VA at the time of DCM diagnosis [14]. In a Danish study on 115 *TTN*tv-related DCM patients with a mean LVEF of 28%, AF and MVA occurred in 43% and 23% of patients, respectively [33]. Of note, AF preceded the DCM diagnosis in 16% of pts, and MVA was the presenting symptom of DCM in 11% [33]. 

In cardiolaminopathies, conduction disease and arrhythmias precede the onset of HF by seven years [34,35,36]. However, *LMNA* mutations are not a common cause of lone AF [37]. Arrhythmias are also reported in *TTN*tv carriers with normal cardiac function. In a case control study that included 2781 participants with early-onset AF and normal LVEF, and 4959 controls, there was a statistically significant association between *TTN*tv and AF [38]. Associations with arrhythmias, including AF, were also observed in the genomics-first study by Haggerty et al., even when conditioning on DCM diagnosis [8]. Among screened relatives in our study, AA or VA was found in 8/50 (16%) subjects and it was the earliest detected abnormality in 6 (12%) of them. This shows that various clinical scenarios are possible in the course of cardiotitinopathies: Arrhythmias appear typically in the late stage of the disease, but they can also precede the DCM diagnosis.

As shown previously by others [7,15] and us [3], LBBB is relatively uncommon among cardiotitinopathy patients and therefore CRT requirement is less pronounced.

Hs-cTnT in clinically stable *TTN*tv-positive DCM patients, although significantly higher in comparison to their non-DCM relatives, was not elevated (median serum level 6.7 ng/L). Cardiotitinopathies are distinguished clearly in this feature from cardiolaminopathies, another relatively common form of inherited DCM, where the hs-cTnT level is already elevated in the preclinical stage of the disease [17]. The Hs-cTnT serum concentration is often elevated in chronic HF. In the meta-analysis of data of patients with chronic HF of different etiologies, Aimo et al. found that hs-cTnT was independently associated with all-cause and cardiovascular mortality [39]. Of note, the median hs-cTnT level in individual cohorts differed significantly. In the largest group of 4053 participants of the Val-HeFT trial, the median hs-cTnT was 12.5 ng/L whereas it was only 4.4 ng/L in the cohort of the VitD-CHF trial and 28.0 ng/L in the study by Nakamura et al. [39] Although this heterogeneity can be explained by factors such as age, severity of HF, and co-morbidities [39], further studies may be needed to highlight the role of other factors, e.g., HF etiology including genetic determinants. It could be highly practical for clinicians to identify genetic factors that lead to early troponin leakage and check whether it is associated with an unfavorable prognosis. Elevated hs-cTnT could be a “red flag” for priority genetic screening of the families and more vigilant follow-up in the patients [40].

In this study, we did not aim to directly compare patients with DCM-causing *TTN* and *LMNA* variants. The vast majority of probands with *LMNA* variants, described in the previous study [17], were not tested with NGS and we cannot exclude the presence of pathogenic variants in other DCM-related genes in them. However, to better illustrate the differences in the course of both forms of inherited DCM, we present selected clinical characteristics and data on the penetration of disease indicators in both groups in Appendix A and Appendix A.

### 4.2. Risk Stratification including Biomarkers

Another major finding of our study was a strong, independent association between the NT-proBNP level ≥650 pg/mL and the occurrence of the composite endpoint of MVA and esHF among *TTN*tv carriers. Recently, several studies defining prognostic factors in cardiotitinopathy have been published; however, none of them included an assessment of circulating biomarkers [7,14,15,16].

An NT-proBNP serum concentration ≥650 pg/mL was also the best predictor of MVA in our study. The excellent prognostic role of NT-proBNP in patients with HF is widely recognized [41]. The association of raised levels of NT-proBNP and MVA in HF patients was shown previously in general HF cohorts [42,43]. NT-proBNP also provides information regarding the risk of SCD in a community-based population beyond other traditional risk factors [44].

There is a great need for the identification of prognostic factors that may help in decision making with regard to ICD therapy. Our study suggests that NT-proBNP, a commonly available circulating biomarker, may be useful in the setting of clinically stable *TTN*tv carriers. 

Hearts of *TTN*tv-positive DCM patients have thinner LV walls and lower indexed LV mass compared to *TTN*tv-negative controls [6], while in arrhythmogenic DCM related to *SCN5A* variants, myocardial thickness is normal [45,46]. It results in higher LV wall stress and release of NT-proBNP and is associated with an increased risk of VA [6,15,47]. It may explain why NT-proBNP may be a good predictor of both esHF and MVA in *TTN*tv-related DCM.

Unlike in cardiolaminopathies, *TTN*tv-positive DCM patients have midwall replacement fibrosis detected in CMR at a similar frequency as *TTN*tv-negative DCM controls [15,47], but interstitial fibrosis is found at endomyocardial biopsy significantly more often [47]. We hypothesize that hs-cTnT leakage, detectable from early stages of cardiolaminopathy [17], may reflect cardiomyocyte death and replacement fibrosis, prevalent in *LMNA*-related cardiac disease, but it may be undetectable in interstitial fibrosis, characteristic of cardiotitinopathy. This might explain why the hs-cTnT level rises significantly only in the end-stage of *TTN*tv-related DCM whilst in earlier stages, when interstitial fibrosis and increased risk of life-threatening arrhythmias are already present, it remains within a normal range. 

### 4.3. Molecular Findings in the Study Cohort

As Giudicessi et al. stated [48], the prevalence of *TTN*tv in the Genome Aggregation Database (1.8%) is more than 4 times higher than the estimated prevalence of DCM in the general population (0.4%). This underlines the role of *TTN*tv as susceptibility variants and suggests that strong environmental effects or additional genetic factors contribute to the development of a cardiac phenotype [49].

In this study, we showed that pathogenic or likely pathogenic *TTN*tv in DCM patients was located in all domains of the gene and had a high proportion spliced-in index, as in the study by Akhtar et al. [50]. Early studies [2], including ours [3], showed the A-band location of *TTN*tv mutations as more specific for DCM patients. With many more *TTN*tv identified and more accumulated data, no statistically significant differences in baseline clinical phenotypes attributable to *TTN*tv location across different TTN bands were found [16].

### 4.4. Study Limitations

This study comes from a tertiary referral center, one of two leading cardiology centers in Poland performing HTx, therefore patients may present with more severe diseases than patients usually admitted in other centers. A major limitation of the study is the small sample size due to its single-center character and, hence, the small number of major cardiovascular events, precluding the use of multivariate analysis models. The retrospective observational design of the study may include confounders. Moreover, because of the retrospective nature of the study, we encountered data gaps that could not be filled. 

## 5. Conclusions

The earliest abnormality emerging in the course of cardiotitinopathies is left ventricular dysfunction, and unlike in cardiolaminopathies, hs-cTnT is not elevated in the early phase of the disease. Therefore, echocardiography cannot be replaced by measurements of circulating cardiac biomarkers in the detection of the onset of the disease in asymptomatic *TTN*tv carriers. An increased NT-proBNP level is the strongest marker of adverse prognosis in *TTN*-related DCM.

## Figures and Tables

**Figure 1 diagnostics-12-00013-f001:**
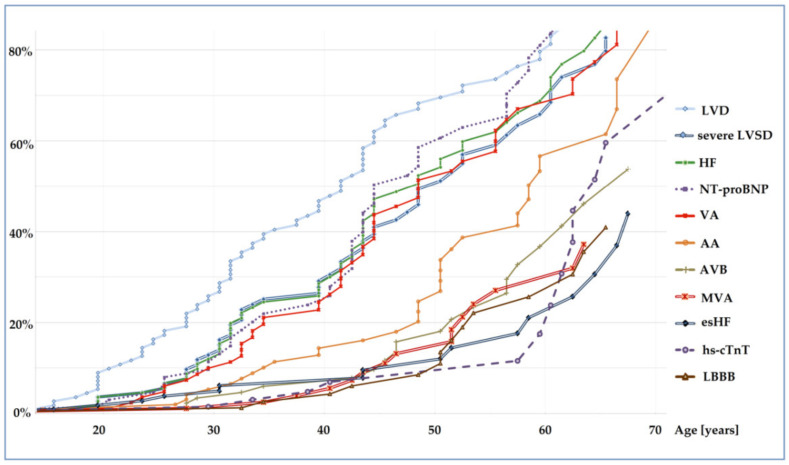
Penetrance of disease indicators estimated by Kaplan–Meier method in the whole study cohort of *TTN*tv carriers. Legend: AA, atrial arrhythmia; AVB, atrioventricular block; esHF, end-stage heart failure; HF, heart failure; hs-cTnT, high-sensitivity cardiac troponin T concentration >14 ng/L; LBBB, left bundle branch block; LVD, left ventricular dysfunction; severe LVSD, severe left ventricular systolic dysfunction; MVA, malignant ventricular arrhythmia; NT-proBNP, N-terminal pro-B-type natriuretic peptide serum concentration >125 pg/mL; *TTN*tv, titin truncating variants; VA, ventricular arrhythmia.

**Figure 2 diagnostics-12-00013-f002:**
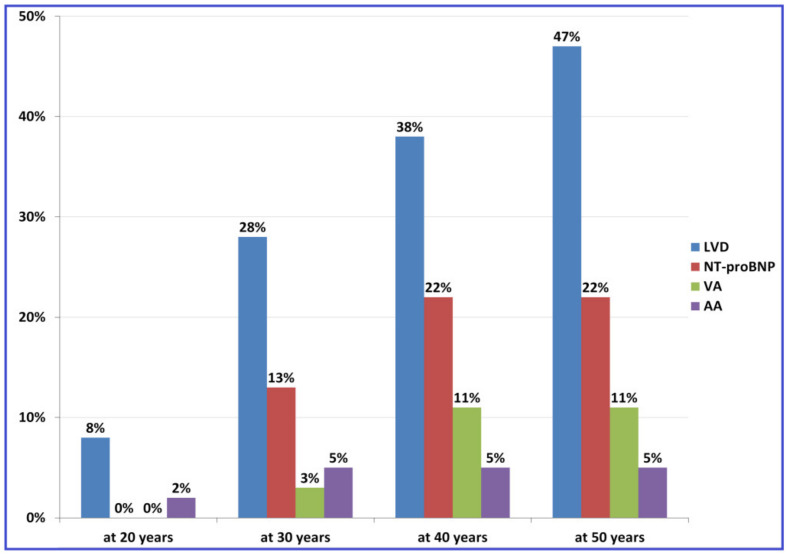
Penetrance of cardiotitinopathy indicators estimated by Kaplan–Meier method in asymptomatic *TTN*tv carriers identified through familial screening. Legend: AA, atrial arrhythmia; LVD, left ventricular dysfunction, NT-proBNP, N-terminal pro-B-type natriuretic peptide serum concentration >125 pg/mL; VA, ventricular arrhythmia.

**Figure 3 diagnostics-12-00013-f003:**
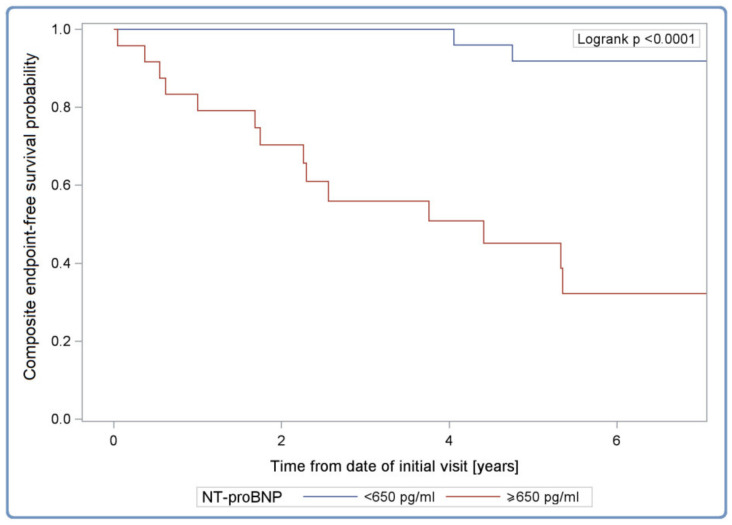
Kaplan–Meier composite endpoint-free survival curves in cardiotitinopathy according to NT-proBNP serum concentration. Legend: NT-proBNP, N-terminal pro-B-type natriuretic peptide serum concentration.

**Figure 4 diagnostics-12-00013-f004:**
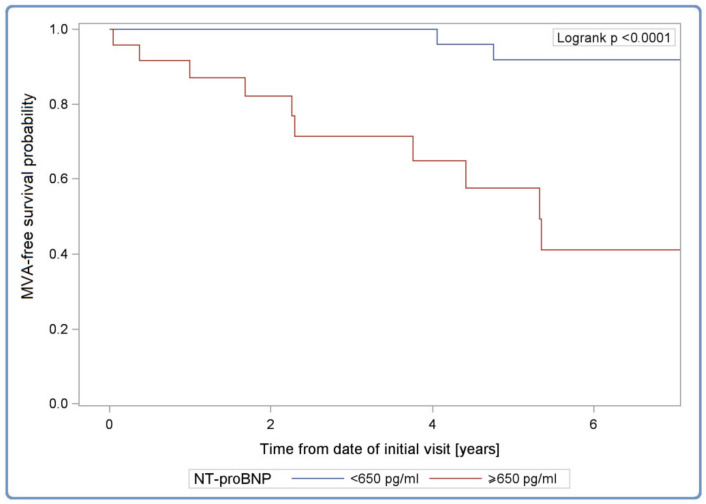
Kaplan–Meier malignant ventricular arrhythmia-free survival curves in cardiotitinopathy according to NT-proBNP serum concentration. Legend: MVA, malignant ventricular arrhythmia; NT-proBNP, N-terminal pro-B-type natriuretic peptide serum concentration.

**Table 1 diagnostics-12-00013-t001:** Baseline clinical characteristics of *TTN* truncating variant carriers.

	All Carriers N = 108	DCM N = 70 (64.8%)	non-DCM N = 38 (35.2%)	*p*
Age, years	39.7 ± 15.5	40.5 ± 14.6	38.3 ± 17.2	0.475
Men	67 (62.0%)	55 (78.6%)	12 (31.6%)	<0.001
Probands	43 (39.8%)	43 (61.4%)	0	<0.001
Symptoms				
Heart failure	54 (50.0%)	54 (77.1%)	0	<0.001
NYHA class ≥3	14 (13.0%)	14 (20.0%)	0	0.002
Family history of SCD <50 years	28 (25.7%)	15 (21.4%)	13 (34.2%)	0.148
Arrhythmias and CCD				
Atrial arrhythmias	25 (23.1%)	22 (31.4%)	3 (7.9%)	0.006
nsVT (n = 106)	42 (39.6%)	38 (55.1%)	4 (10.8%)	<0.001
LBBB	11 (10.2%)	11 (15.7%)	0	0.007
AV block (≥1st degree)	15 (13.9%)	13 (18.6%)	2 (5.3%)	0.056
Echocardiography				
LVEF < 50%	66 (61.1%)	62 (88.6%)	4 (10.5%)	<0.001
LVEF, %	43.5 ± 13.8	36.2 ± 11.0	56.9 ± 6.6	<0.001
LVEDD, mm	58.3 ± 9.6	63.2 ± 8.0	49.4 ± 4.8	<0.001
LAs, mm (n = 103)	40.6 ± 8.1	43.7 ± 8.0	35.3 ± 4.9	<0.001
Biomarkers in stable phase				
hs-cTnT, ng/L (n = 90)	4.4 [<3.0; 8.3]	6.7 [3.8; 9.3]	<3.0 [<3.0; 4.2]	<0.001
hs-cTnT > 14 ng/L	9 (10.0%)	7 (12.5%)	2 (5.9%)	0.474
NT-proBNP, pg/mL (n = 72)	244 [76; 1225]	534 [157; 1498]	72 [23; 94]	<0.001
NT-proBNP > 125 pg/mL	47 (65.3%)	46 (79.3%)	1 (7.1%)	<0.001
NT-proBNP > 650 pg/mL	24 (33.3%)	24 (41.4%)	0	0.003
Implantable devices				
PM for bradyarrhythmias	5 (4.6%)	5 (7.1%)	0	0.159
CRT-D	2 (1.8%)	2 (2.9%)	0	0.540
ICD/CRT-D	14 (13.0%)	14 (20.0%)	0	0.002

Legend: Number of subjects is expressed as n (%). Continuous variables are shown as mean ± standard deviation or median and quartiles [Q1:25th- Q2:75th percentiles]. AV block, atrioventricular block; CCD, cardiac conduction defect; CRT-D, cardiac resynchronization therapy defibrillator; DCM, dilated cardiomyopathy; HF, heart failure; hs-cTnT, high-sensitivity cardiac troponin T serum concentration; ICD, implantable cardioverter defibrillator; LAs, left atrial systolic dimension; LBBB, left bundle branch block; LVEDD, left ventricular end-diastolic dimension; LVEF, left ventricular ejection fraction; nsVT, non-sustained ventricular tachycardia; NT-proBNP, N-terminal pro-B-type natriuretic peptide serum concentration; NYHA class, New York Heart Association functional class; PM, pacemaker; SCD, sudden cardiac death.

**Table 2 diagnostics-12-00013-t002:** Clinical outcomes in the cohort of *TTN* truncating variant carriers.

Events during Follow-Up	Total N = 108	Men N = 67 (62.0%)	Women N = 41 (38.0%)	*p*
ICD in secondary prophylaxis	2 (1.9%)	1 (1.5%)	1 (2.4%)	1.00
CRT-D	7 (6.5%)	6 (9.0%)	1 (2.4%)	0.249
ICD/CRT-D implantation	27 (25.0%)	21 (31.3%)	6 (14.6%)	0.052
Malignant ventricular arrhythmia, n = 107	13 (12.1%)	9 (13.4%)	4 (10.0%)	0.763
Appropriate ICD intervention, n = 27	13 (48.1%)	9 (42.9%)	4 (66.7%)	0.384
Cardiopulmonary resuscitation, n = 106	2 (1.9%)	2 (3.0%)	0	0.530
Sudden cardiac death, n =106	1 (0.9%)	0	1 (2.6%)	0.368
End-stage heart failure, n = 107	13 (12.1%)	12 (17.9%)	1 (2.5%)	0.024
LVAD	5 (4.6%)	5 (7.3%)	0	0.155
Heart transplantation	8 (7.4%)	8 (11.9%)	0	0.024
HF death, n = 106	5 (4.6%)	4 (3.7%)	1 (2.6%)	0.650
Death	9 (8.3%)	6 (9.0%)	3 (7.3%)	1.00

Legend: Number of subjects with events is expressed as n (%). CRT-D, cardiac resynchronization therapy defibrillator; HF, heart failure; ICD, implantable cardioverter defibrillator; LVAD, left ventricular assist device.

**Table 3 diagnostics-12-00013-t003:** Potential risk factors affecting occurrence of the composite endpoint of malignant ventricular arrhythmia and end-stage heart failure in cardiotitinopathy.

	Cumulate Incidence	*p*-Value Log-Rank	Univariable		Multivariable	
			HR [95% CI]	*p*-Value Wald	HR [95% CI]	*p*-Value Wald
MVA + esHF	at 6 Years of Follow-Up					
Sex: male vs. female	32 vs. 14	0.033	3.08 [1.04; 9.18]	0.043		
AA: yes vs. no	50 vs. 18	0.001	3.77 [1.56; 8.90]	0.002		
nsVT: yes vs. no	46 vs. 7	<0.001	5.5 [1.9; 16.5]	0.002		
LAs: ≥45 vs. <45 mm	67 vs. 5	<0.001	28.9 [6.7; 124.8]	<0.001		
LVEF: <30 vs. ≥30%	78 vs. 9	<0.001	14.0 [5.5; 35.7]	<0.001		
LBBB: yes vs. no	86 vs. 16	<0.001	8.5 [3.6; 20.4]	<0.001		
NT-proBNP ≥650 vs. <650 pg/mL	68 vs. 8	<0.001	14.4 [3.6; 57.5]	<0.001	31.3 [4.0; 246] **^#^**	0.001 **^#^**
hs-cTnT: ≥18 vs. <18 ng/L	75 vs. 13	<0.001	7.7 [2.1; 28.6]	0.002		

Legend: AA, atrial arrhythmia; CI, confidence interval; esHF, end-stage heart failure; HF, heart failure; HR, hazard ratio; hs-cTnT, high-sensitivity cardiac troponin T serum concentration; LAs, left atrial systolic dimension; LBBB, left bundle branch block; LVEF, left ventricular ejection fraction; MVA, malignant ventricular arrhythmia; nsVT, non-sustained ventricular tachycardia; NT-proBNP, N-terminal pro-B-type natriuretic peptide serum concentration; ^#^, adjusted for intra-family correlations.

**Table 4 diagnostics-12-00013-t004:** Potential risk factors affecting occurrence of malignant ventricular arrhythmia in cardiotitinopathy.

	Cumulate Incidence	*p*-Value Log-Rank	Univariable		Multivariable	
			HR [95% CI]	*p*-Value Wald	HR [95% CI]	*p*-Value Wald
MVA	at 6 Years of Follow-Up					
Sex: Male vs. Female	21 vs. 14	0.362	1.72 [0.52; 5.59]	0.368		
SCD <50 in family: yes vs. no	13 vs. 20	0.684	0.76 [0.21; 2.78]	0.685		
AA: yes vs. no	35 vs. 14	0.010	3.81 [1.28; 11.36]	0.016		
nsVT: yes vs. no *	39 vs. 2	<0.001	12.5 [2.2; 72.7]	0.005		
LAs: ≥45 vs. <45 mm	52 vs. 5	<0.001	16.9 [3.7; 77.4]	<0.001		
LVEF: <30% vs. ≥30%	65 vs. 8	<0.001	9.9 [3.2; 30.2]	<0.001		
LBBB: yes vs. no *	83 vs. 10	<0.001	14.6 [4.8; 44.3]	<0.001		
NT-proBNP ≥650 vs. <650 pg/mL	59 vs. 8	<0.001	12.7 [2.8; 58.3]	0.001	11.7 [2.4; 56.6] **^#^**	0.002 **^#^**
hs-cTnT: ≥14 vs. <14 ng/L	25 vs. 13	0.808	1.29 [0.16; 10.3]	0.808		

Legend: AA, atrial arrhythmia; CI, confidence interval; HF, heart failure; HR, hazard ratio; hs-cTnT, high-sensitivity cardiac troponin T serum concentration; LAs, left atrial systolic dimension; LBBB, left bundle branch block; LVEF, left ventricular ejection fraction; MVA, malignant ventricular arrhythmia; nsVT, non-sustained ventricular tachycardia; NT-proBNP, N-terminal pro-B-type natriuretic peptide serum concentration; SCD, sudden cardiac death; *, Firth’s correction; ^#^, adjusted for intra-family correlations.

## Data Availability

The data presented in this study are available on request from the corresponding author (Z.T.B.). The data are not publicly available due to privacy concerns.

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
