# Peer review of "Titin-Related Dilated Cardiomyopathy: The Clinical Trajectory and the Role of Circulating Biomarkers in the Clinical Assessment"

_diagnostics, 2021, doi:10.3390/diagnostics12010013_

Round 1

Reviewer 1 Report

Significant work. Interesting results. Like you already say is to be extendend in a multicenter analysis.

Great work.

Congratulation

Author Response

Thank you very much for your positive review.

Reviewer 2 Report

The work proposed by Chmielewski et al. is of great interest for those involved in cardiomyopathies and, more generally, for cardiovascular genetics. The dialted cardiomyopathies (DCM) due to Titin truncating mutations / variants (TTNtv) represent about 20-25% of the total DCM (at least for the Caucasian population). The advent of Next Generation Sequencing has allowed a more systematic molecular approach in the analysis of the causative gene of this disease (Titin gene spans for 108 kb for more than 300 coding exons) and consequently a more accurate genotype-phenotype analysis.

This work analyzes a court of 108 subjects with DCM belonging to a single heart transplant center; 41 TTNtv were identified in 41 of these patients. The main objective is the identification of biomarkers that can predict the prognostic outcomes of patients with DCM carrying TTNtv.

The search for biomarkers (whether they are serum as well as instrumental such as cardiac imaging or cardiac assessments) is of fundamental importance for the implementation of personalized medicine and in the context of TTNtv DCM this information is currently lacking. Consequently, this work fills an urgent clinical need.

I would improve the following aspects:

  1. In the genotyping table I would list the proband (proband 1, 2, 7, 14 etc etc) with the relative mutation or possibly concomitant other mutations in other genes as mentioned in section 3.1
  2. I would make a table / figure in which the differences between DCM caused by LMNA / C and TTN are summarized for an easier usability by the reader

Author Response

We thank you kindly for your efforts to review our manuscript and for your valuable suggestions.

Responding to the issues raised:

"1. In the genotyping table I would list the proband (proband 1, 2, 7, 14 etc etc) with the relative mutation or possibly concomitant other mutations in other genes as mentioned in section 3.1"

Thank  you for highlighting this.

We changed the Table S1 accordingly, to list individual  probands instead of variants. The carriers of the variants who were excluded from the study due to identification of likely pathogenic variants in other DCM-related genes were marked with an asterisk (*) which could be difficult to notice – to make it clearer we used the colour orange in the modified table. However, we did not add another column for concomitant mutations to the Table S1– it would be populated in only three of 46 lines.

Instead, we added the probands' numbers and corresponding TTN variants to the Table S2.

"2. I would make a table / figure in which the differences between DCM caused by LMNA / C and TTN are summarized for an easier usability by the reader."

Our response:

Thank you for bringing this up.

In fact, in this study, we did not aim to directly compare patients with DCM-causing TTN and LMNA variants. The main obstacle was that the vast majority of probands with LMNA variants described in our previous study were tested only with Sanger and not with NGS, and we cannot exclude the presence of pathogenic variants in other DCM-related genes in them. This could be a reason for contesting the results of the study.

However, we agree that the differences in the course of TTN-related and LMNA-related DCM are worth highlighting. We prepared the baseline clinical characteristics of both cohorts in the supplementary Table S5 and the comparison of penetrance of selected disease indicators in the supplementary Figure S1, and added a passage in the discussion (section 4.1) referring to these data.